# Deformation Behavior under Tension with Pulse Current of Ultrafine-Grain and Coarse-Grain CP Titanium

**DOI:** 10.3390/ma16010191

**Published:** 2022-12-25

**Authors:** Vladimir Stolyarov, Oleg Korolkov, Alexander Pesin, George Raab

**Affiliations:** 1Mechanical Engineering Research Institute of Russian Academy of Sciences, 101990 Moscow, Russia; 2Department of Materials Processing Technologies, Nosov Magnitogorsk State Technical University, 455000 Magnitogorsk, Russia

**Keywords:** tension, pulse current, electroplastic effect, titanium, stress, grain size

## Abstract

The problem of the real existence of the electroplastic effect during deformation of metallic materials of different nature is still relevant. At the same time, the influence of structure refinement is not considered enough. In this work, the deformation behavior of ultrafine-grained (UFG) titanium Grade 4 is compared with that of coarse-grained (CG) titanium under tension with pulse current of the low duty cycle. The deformation curves of both structure states are presented for different regimes of pulsed current and thermal heating from an external source. Structure studies by optical and scanning electron microscopy, as well as microhardness measurements have been carried out. It is shown that Grade 4 titanium under tension accompanied by pulsed current exhibits electroplastic effect (EPE) in the form of a flow stress reduction. EPE in UFG state is much stronger than in CG state. An increase in the density and duration of the current pulse leads to a multiple decrease in the flow stresses in CG and UFG titanium. The contribution in the flow stress reduction from heating by an external source was less than that from tension with pulse current at the same temperatures. The impact of pulsed current during tension does not influence microhardness and grain size.

## 1. Introduction

It is well known that electric current in solid metallic materials leads to a number of physical phenomena. The best-known phenomenon is the thermal effect of the current, which causes the temperature to rise according to the Joule–Lenz law. If the electric current is pulsed or alternating, then, due to electromagnetic induction, additional current phenomena arise such as vibration, spin or pinch effects. [1,2]. Their influence leads to a non-uniform current distribution over the conductor cross section and the appearance of mechanical stresses and strains, the magnitude of which largely depends on the current frequency. In the case of combining electric current and plastic deformation of the conductor, the so-called electroplastic (EPE) and magnetoplastic (MPE) effects can be observed, which can cause a change in the material flow stresses. For example, in original articles and numerous reviews, it was shown that at a sufficiently high current density (more than 10^3^–10^4^ A/mm^2^) the electroplastic effect in pure coarse-grained metals and alloys increases deformability and reduces flow stresses by tens of percent [3,4]. 

Another manifestation of EPE is stress jumps directed downwards and observed on the tensile curves at short pulse duration of 10^−5^–10^−4^ s^−1^ and a current density above the critical one. Thus, in the first studies of EPE in Zn, Cd, Sn, Pb, and In single- and polycrystals under tension and compression in the region of plastic flow, downward stress jumps are demonstrated, the amplitude of which increases with the current density [5]. It was noted that the amplitude of jumps in polycrystals was smaller than in single crystals. Later, for a Zn single crystal, it was shown that an increase in the frequency of the pulsed current by two orders of magnitude led to a threefold decrease in flow stresses and a twofold increase in plasticity before failure [6]. In titanium and aluminum polycrystals, the EPE was noticeably lower, although it amounted to tens of MPa [7,8]. The amplitude of stress jumps can reach tens and hundreds of MPa, which depends on the material, its electrical and thermal conductivity, and pulse current parameters, among which the current density, pulse duration and frequency are the most important [9]. These current parameters often determine the ratio of thermal and electroplastic effects, as well as conclusions about the EPE mechanism [3]. Here, we note that the attention of most researchers to the current pulse frequency during tension test was noticeably lesser than to the current density and pulse duration. Mention of the influence of the pulse current frequency on mechanical properties during tension was discovered only in one monograph [10]. As a rule, for practical applications, it is preferable to use higher frequencies (low duty cycle) [11].

The possibility of practical application of EPE in different materials is considered in [10]. Of particular interest is the use of EPE for the deformation of titanium and alloys based on it. Due to the excellent set of physical and mechanical properties, these materials are widely used in many industries, including medical implants [12]. A feature of the deformation of titanium and its alloys is the use of hot rolling. However, in this case, oxides and an alpha layer are formed on the surface of the semi-finished product, with simultaneous grain growth. At the same time, plastic deformation of workpieces without the use of heating is often accompanied by cracking and high springback. Therefore, deformation using EPE at lower temperatures can be a solution for this problem.

It was mentioned above that the nature of the material is important in the implementation of EPE. The literature data indicate the undoubted existence of EPE in pure titanium and its absence in copper and iron [9,13]. Another important aspect when considering the nature of EPE is the microstructure of the material under study, in particular, the grain size. As it turned out, many researchers considered only coarse-grained (CG) metals and alloys [3], while the manifestation of EPE in ultrafine-grained (UFG) and nanocrystalline materials was not studied. The reason for this was the lack of a well-developed and certified method for obtaining UFG structures in bulk samples. It was important to understand the role of grain size in the development of EPE since a large volume fraction of grain boundaries and, accordingly, the density of defects in them can significantly affect the ratio of thermal and electroplastic effects. For the first time, the grain size effect on EPE in commercial pure Ti was demonstrated in [14]. It turned out that under the action of single current pulses of low frequency with a density of 1500–3000 A/mm^2^ and a pulse duration of 100–1000 μs, the amplitude of stress jumps on the loading curve decreases many times as the grain size decreases by two orders of magnitude. However, the performed study concerned only the amplitude of stress jumps under pulse current of high duty cycle, and not a comparison of the reduction in flow stresses in CG and UFG titanium.

In this regard, the aim of the work is to study the effect of low duty cycle pulsed current modes during tension on the microstructure, microhardness, and the relative contribution to the reduction in flow stresses from thermal and electroplastic effects in CG and UFG titanium. 

## 2. Materials and Methods

The research material was commercially pure Grade 4 titanium in two initial states: a—CG (delivery according to ASTM F67-06), in the form of a rod ∅6 mm; b—UFG obtained by the equal channel angular pressing (ECAP)-CONFORM method and subsequent annealing at 300 °C [15]. The microstructure of the samples in the initial (before tension) states is shown in Figure 1. 

The chemical composition of the material specified in the supplier’s certificate is shown in Table 1.

Cylindrical tensile specimens were cut in the longitudinal direction of the bar; the shape and dimensions of the tensile specimens are shown in Figure 2.

Tension test was performed on a horizontal tensile testing machine IR-5081/20. The test speed was 1 mm/min. A pulsed current was supplied from the generator to the clamps of the tensile testing machine. To compare the contributions of the thermal and electroplastic effects at the same temperatures, one of the samples was heated during tension with air from a MAKITA HG6530VK hot air gun (dryer). Used tension and current modes are presented in Table 2. 

The duty cycle of the pulsed current was q = T/τ = 1/ντ, where T, τ and ν period, pulse duration and frequency, respectively. The parameters τ and ν were varied so that the duty cycle was constant q = 10. The sample grippers were isolated from the machine by fiberglass spacers. The temperature of the sample during tension was controlled by a device Digital Thermometers UT320 Series and chromel–alumel thermocouple in the center of the sample with accuracy of ±2 °C. The amplitude current density was monitored using an AKIM-4131/2 oscilloscope. The scheme of the test stand is shown in Figure 3.

The microhardness and optical microstructure of the samples in the deformed and undeformed zones in the longitudinal section were studied in accordance with the scheme (Figure 4), respectively, on the PMT-3 device (at a load of 100 g, exposure time 15 s, at least ten measurements per point, measurement accuracy ± 5%) and optical microscope Olympus Bx-51.

The preparation of samples for microhardness measurements and optical microscopy was carried out sequentially by mechanical grinding and polishing, then by electropolishing at a voltage of 35–50 V in a solution of perchloric and acetic glacial acids in a ratio of 1:4. The microstructure was prepared by chemical etching in a solution of hydrofluoric and nitric acids in water in a ratio of 1:2:47 parts. 

The microstructure of the thin layers of the samples was analyzed using a JEM 2100 transmission electron microscope. The blanks for the foils were cut by the electro erosion method and mechanically thinned to a thickness of 100 µm using abrasive paper. Then, foils were prepared by double-sided electrolytic polishing on TenuPol-5 (Struers LLC, Cleveland, OH, USA) in a solution of 5% perchloric acid, 35% butanol and 60% methanol at −30 °C. Fractographic images were obtained using a Tescan Mira 3 LMU scanning electron microscope (Tescan, Brno, Czechia). For better contrast, the samples were mounted at an angle of 45° to the direction of the electron beam.

## 3. Results

### 3.1. Microstructure and Microhardness

Figure 5 and Figure 6 show microstructures of CG titanium in different zones of the sample, tested with the maximum current density and heating with a dryer. In general, the microstructure reflects a partially recrystallized state with almost equiaxed grains with an average size of 13 μm. It can be seen that the shape and size of the grains with the introduction of a pulsed current with a maximum density and duration (Figure 5) or heating with a dryer (Figure 6) practically do not change along the length of the sample. However, in the neck region, the grains are elongated in the direction of tension and the appearance, mainly inside the grains, of particles of the second phases, which increases with increasing pulse duration in the neck region. An EDX analysis of single particles by an attachment to an electron microscope Tescan Mira 3 LMU showed that their composition corresponds toTiO_2_ (Table 3). 

Figure 7 shows the dependences of the microhardness of CG and UFG titanium on the test modes and the place of measurement. 

It can be seen that for both states, tension without current and with current increases the microhardness in the neck compared to the undeformed zone of the sample by 25 and 15%, respectively, for CG and UFG titanium. Heating with a dryer does not affect the microhardness. In UFG titanium, the microhardness decreases in the deformable zone, which is not observed in CG titanium.

### 3.2. Tension

The stress-strain curves for CG and UFG titanium at various test regimes are presented in Figure 8. 

In CG titanium, when a pulsed current with a density of 30 A/mm^2^ (curves 2, 3) and 60 A/mm^2^ (curves 4, 5) is introduced, the flow stress decreases by 10–50% compared to tension without current (curve 1), which increases with increasing density and pulse duration (Figure 8a). The higher the current density, the stronger the influence of the pulse duration. At a current density of 60 A/mm^2^, the strain hardening noticeably decreases. Heating with a dryer (curve 6) also reduces the flow stresses, but noticeably less than when tested with current at the same temperatures. Heating with a dryer and the introduction of current contribute to a slight increase of 2–4% elongation to failure, which is more pronounced at maximum current density. For all tension conditions, the angle of inclination (modulus of elasticity) in the elastic area noticeably decreases with increasing current regimes and when heated with a dryer. 

In UFG titanium, the changes listed above are qualitatively similar, but quantitatively more pronounced than in CG titanium (Figure 8b). For example, an increase in the current density and pulse duration contributes to a greater decrease in the flow stress (up to 82%), leads to the disappearance of the yield plateau and the appearance of a yield tooth (Figure 8b, curve 5 and inset). The ratio of uniform and concentrated deformation is shifted towards stronger necking. The relative contribution to the reduction in flow stresses at the same temperatures from the introduction of current increases in comparison with the contribution from heating with a dryer (Figure 8b, curve 6). 

The mechanical properties of CG and UFG titanium for all the above modes are given in Table 4.

### 3.3. Fracture

Figure 9 shows fractographic images of the fracture zone of the samples. 

The view of the surface failure for CG titanium corresponds to a ductile fracture; ridges and pits of dimples with a size of 10–20 μm are visible, as well as single pores that make up no more than 2% of the area under consideration (Figure 9a). The size of the dimples correlates with the grain size in CG titanium. The introduction of a pulsed current did not lead to a noticeable change in the nature of fracture; however, the dimple size slightly decreased (Figure 9b), and the content of pores did not change.

The nature of the fracture for UFG titanium also corresponds to a ductile fracture (Figure 9b); ridges and large dimples 2–3 μm in size are visible, inside which smaller dimples of up to 500 nm in size are observed. The introduction of a pulsed current did not lead to a change in the nature of the fracture.

## 4. Discussion

Structural studies of samples after different types of tests showed no changes in the grain size for CG titanium, but found the appearance of particles of second phases, mainly in the neck of the samples. This is confirmed by measurements of microhardness, which increased only in the neck of the sample and was due to particles of the second phase. In UFG titanium, the thermal effect of the current in the deformable zone leads to stress relaxation (decrease in microhardness) without changing the grain size. The results of UFG titanium fractography also indirectly indicate the absence of grain growth. That is, the relatively low temperature caused by the thermal effect of the current could not cause grain growth in both structural states. Despite this, all tension curves for CG and UFG titanium under current or heating by an external source demonstrate a significant decrease in the elastic modulus, flow stresses, tensile strength, uniform deformation, and a slight increase in relative elongation. The reduction in tensile strength and yield stress at the same temperatures and no grain growth for testing with current is almost twice as large as when heated with a dryer. This fact is one of the signs of the manifestation of nonthermal EPE by titanium, regardless of its grain size. Previously, on the same Grade 4 titanium, the manifestation of EPE was shown not only in tension, but also in bending [16]. Qualitatively similar results were obtained for the commercial pure and less contaminated Grade 2 titanium in [9]. However, quantitative changes in the present study were more significant. A decrease by two orders of magnitude of the grain size in the original titanium leads to a stronger effect of EPE on the reduction in flow stresses in UFG titanium (82%) compared to CG titanium (65%). This effect is possibly associated with a higher density of grain boundaries and greater local heating in UFG titanium. Interestingly, the effect of structural refinement on the reduction of flow stresses in CG and UFG titanium is opposite to the effect of reduction in the amplitude of stress jumps described in [14]. It is assumed that this is due to different mechanisms of plastic deformation under the action of current inside individual grains and in the bulk of the sample. The amplitude of the stress jump from a single pulse is related to the free path of single dislocations, which decreases with decreasing grain size. The decrease in flow stresses at a low duty cycle current depends on the volume fraction of grain boundaries, which increases with a decrease in grain size. As for the decrease in the modulus of elasticity, since it is observed with the introduction of current and heating with a dryer, the mechanism of the “electronic wind” cannot be the main reason here. It is possible that the decrease in the modulus of elasticity under current is associated with the additional formation of mobile dislocations in the body of grains or vacancies, pores, and microcracks at the grain boundaries [17]. The strong localization of deformation in the neck of the samples, which is visible in the current-induced tension curves, is confirmed by an increase in microhardness in this region. The physical reason for the appearance of a yield “tooth” in UFG titanium during tension with an intense current regime may be the migration of oxygen atoms and the formation of Cottrell atmospheres [18]. 

## 5. Conclusions

The effect of electrical current of low duty cycle on the microstructure, microhardness and deformation behavior under tension of titanium Grade 4 in course-grained and ultrafine-grained states was investigated by changing current density and pulse duration, as well as by comparison with tension under heating. Based on the results, the following conclusions could be drawn:
It was shown that the tension of Grade 4 titanium specimens accompanied by a low duty cycle pulsed current leads to EPE, which manifests itself in a decrease in flow stresses. In this case, the maximum relative effect of reducing the flow stresses in UFG more than in CG titanium, 82% and 65%, respectively.An increase in the density and duration of the current pulse leads to a multiple decrease in the flow stresses in CG and UFG titanium.EPE in CG and UFG titanium under the influence of the selected current modes manifests itself in a strong decrease in flow stresses without a significant change in microhardness and grain size.Based on these conclusions, future research should be focused on cold rolling of UFG titanium accompanied by pulsed current according to the found modes and related microstructural observations and mechanical tests.

## Figures and Tables

**Figure 1 materials-16-00191-f001:**
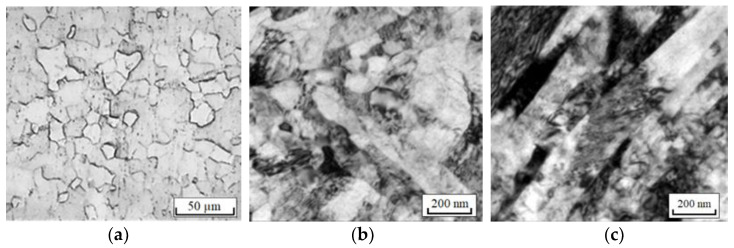
Microstructure of titanium in CG (**a**) and UFG (**b**,**c**) states: (**a**)—along; (**b**)—cross section; (**c**)—longitudinal section.

**Figure 2 materials-16-00191-f002:**
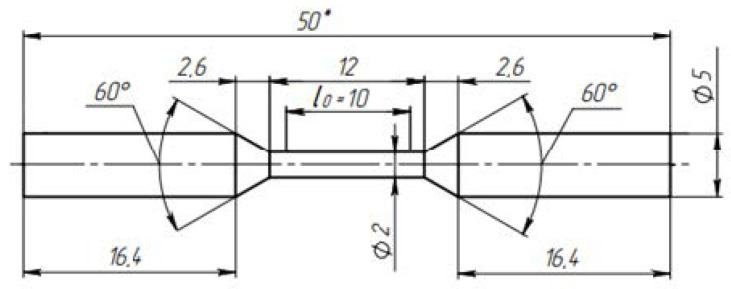
Shape and dimensions of tensile specimens.

**Figure 3 materials-16-00191-f003:**
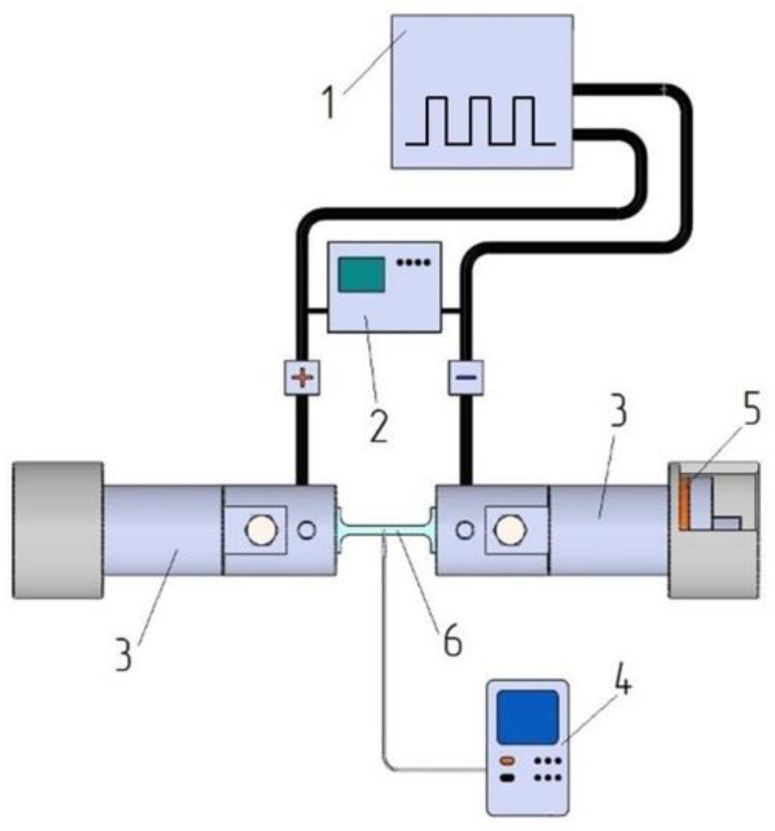
Diagram of the test setup: 1—pulse current generator; 2—oscilloscope; 3—captures; 4—thermocouple; 5—isolation; 6—sample.

**Figure 4 materials-16-00191-f004:**
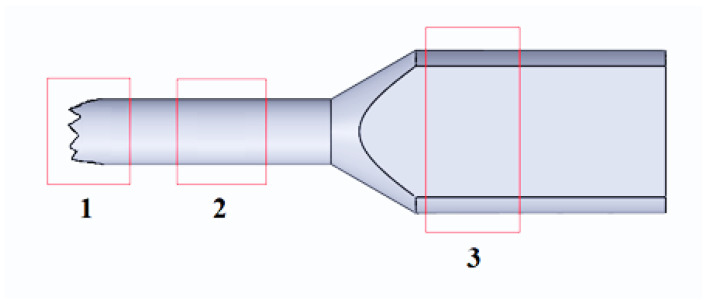
Scheme of places for measuring microhardness and studying microstructure: 1—neck; 2—deformable zone; 3—non-deformable zone.

**Figure 5 materials-16-00191-f005:**
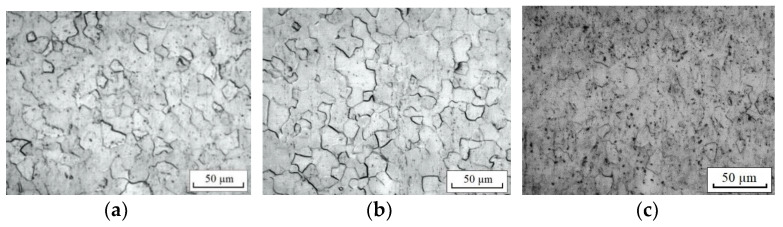
Microstructure of CG titanium tested with current (60 A/mm^2^, 500 μs, 200 °C): (**a**)—undeformed zone; (**b**)—deformed zone; (**c**)—neck.

**Figure 6 materials-16-00191-f006:**
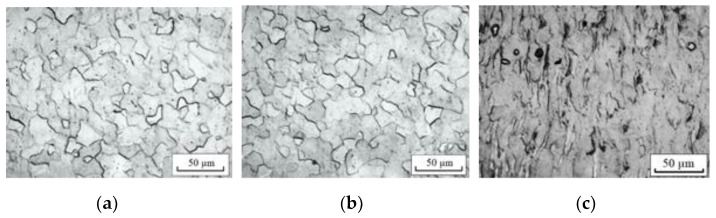
Microstructure of CG titanium tested by heating with a dryer, 200 °C: (**a**)—undeformed zone; (**b**)—deformed zone; (**c**)—neck.

**Figure 7 materials-16-00191-f007:**
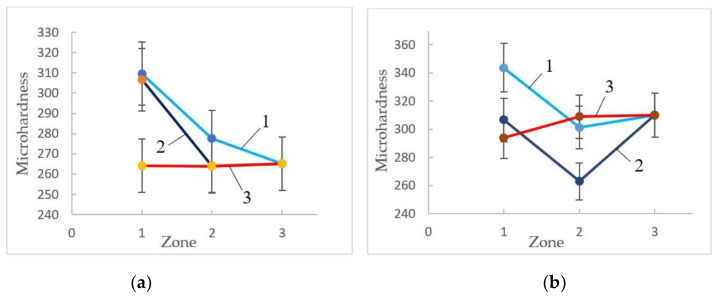
Changes in microhardness along the sample length in (**a**) CG and (**b**) UFG titanium depending on tension conditions: 1—without current; 2—j = 60 A/mm^2^, τ = 500 µs; 3—dryer, 200 °C.

**Figure 8 materials-16-00191-f008:**
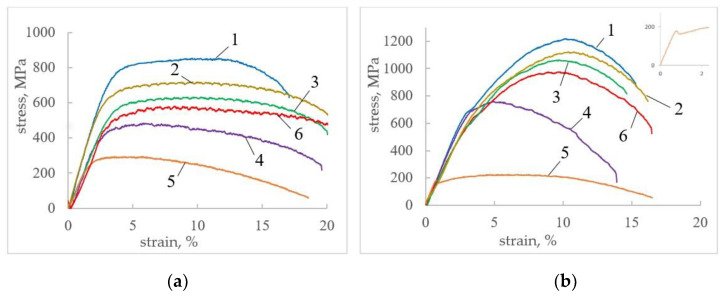
Stress-strain curves for CG (**a**) and UFG (**b**) titanium: 1—without current; 2—j = 30 A/mm^2^, τ = 100 µs; 3—j = 30 A/mm^2^, τ = 1000 μs; 4—j = 60 A/mm^2^, τ = 100 μs.; 5—j = 60 A/mm^2^, τ = 500 μs; 6—dryer, 200 °C; Inset: Yield tooth on curve 5.

**Figure 9 materials-16-00191-f009:**
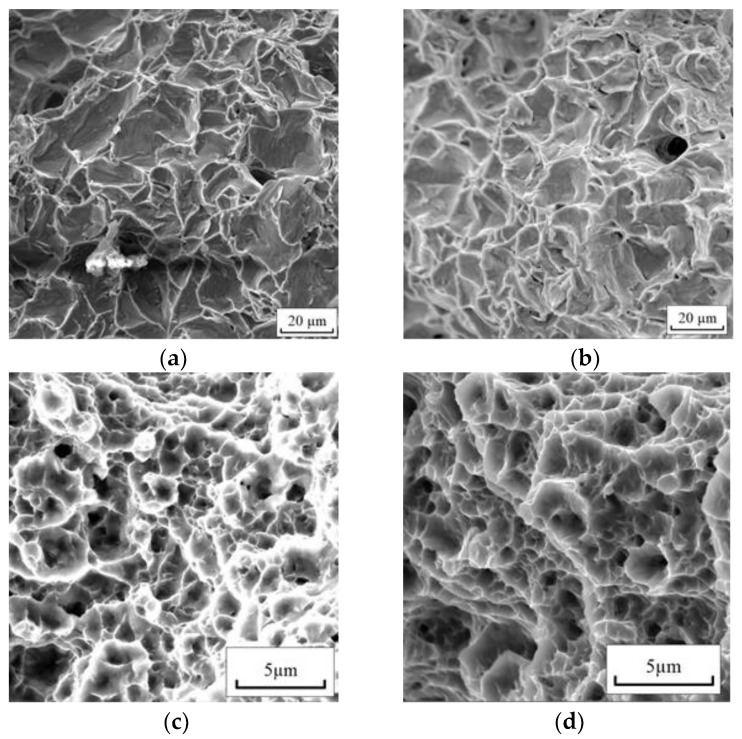
Fracture surface of CG (**a**,**b**) and UFG (**c**,**d**) Grade 4 under tension: (**a**,**c**)—no current; (**b**,**d**)—with current of j = 60 A/mm^2^ τ = 100 μs.

**Table 1 materials-16-00191-t001:** Chemical composition of Grade 4 (wt.%).

N	C	H	Fe	O	Ti
0.01	0.04	0.015	0.14	0.36	balance

**Table 2 materials-16-00191-t002:** Experimental modes for CG and UFG Ti.

No.	State	Method of Action	Current Density, A/mm^2^	Pulse Duration, μs
1	CG	Pulse current	-	-
2			30	100
3				1000
4			60	100
5				500
6		Dryer 200 °C	-	-
1	UFG	Pulse current	-	-
2			30	100
3				1000
4			60	100
5				500
6		Dryer 200 °C	-	-

**Table 3 materials-16-00191-t003:** The element content in the particles (wt.%).

Spectrum Label	Sample Center
O	4.52
Ti	95.27
Balance	0.21
Total	100

**Table 4 materials-16-00191-t004:** Mechanical properties of CG and UFG titanium.

No.	State	Method of Action	j, A/mm^2^	*τ*, μs	T, °C	UTS, MPa	YS, MPa	*δ*, %	Δσ *^,^ MPa	Δσ, %
1	CG	No current			25	850	610	15		
2		Pulse current	30	100	40	720	520	18	130	15
3				1000	40	630	515	18	220	26
4			60	100	110	480	370	19	370	44
5				500	200	295	265	19	555	65
6		Dryer			200	575	495	18	275	32
1	UFG	No current			25	1215	650	12		
2		Pulse current	30	100	40	1120	660	13	95	8
3				1000	40	1060	515	11	155	13
4			60	100	120	760	630	13	455	38
5				500	200	220	170	16	995	82
6		Dryer			200	970	560	14	245	20

*—Δσ = UTS_(no current)_ − UTS _(with current or dryer)_.

## Data Availability

The data used to support the findings of this study are included within the article.

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
