# Peer review of "Deformation Behavior under Tension with Pulse Current of Ultrafine-Grain and Coarse-Grain CP Titanium"

_materials, 2022, doi:10.3390/ma16010191_

Round 1

Reviewer 1 Report

Electroplastic effec  for UFC and CG commercial pure Titanium was studied and compared. However, the novelty is not enough. As the EPE had been studied by the author for UFC titanium alloys ten years ago. This study gave no new finds or understanding of the phenomenon. It is suggested to give a thorough analysis about the mechanism of the difference on EPE of UFC CP Ti and CG CP Ti, and with a comprasion with pre studied titanium alloys.

the main problem of the paper is similar results had been published by the author for Titanium alloy which is not listed in the references and no comparison was did with this research. I recommend reject, and if the author want to improve the paper, it is sugguest to give a through analysis about the mechanism for EFP effect of CP Ti, if it is the same or different with the pre-published titanium alloy.    

Reviewer 2 Report

Dear authors,

 the paper contains interesting analysis. However, here are some suggestions to improve the manuscript:

L 51-53: “Here we note that…” please provide adequate references for this statement

Please describe the origin of Figure 1. If it is your work, please explain the metallographic preparation, the equipment used, the conditions, the microscope, etc.

Also clarify where the data listed in Table 1 come from. Did you get them from the manufacturer or did you do the analysis yourself? If you did the analysis yourself, describe which ones, how, etc

L 155: “..electron microscope Tescan Mira 3 LMU showed..” you should add numerical data to prove this statement. Also edx spectruum would be useful.

L 214: 3.3. Instead Fructure it should be Fracture

Based on conclusions, please outline a possible future research.

Round 2

Reviewer 1 Report

The authors try to modify the manuscript and the response seems fine.